# Cellulose Nanostructure-Based Biodegradable Nanocomposite Foams: A Brief Overview on the Recent Advancements and Perspectives

**DOI:** 10.3390/polym11081270

**Published:** 2019-07-31

**Authors:** Mpho Phillip Motloung, Vincent Ojijo, Jayita Bandyopadhyay, Suprakas Sinha Ray

**Affiliations:** 1DST-CSIR National Centre for Nanostructured Materials, Council for Scientific and, Industrial Research, Pretoria 0001, South Africa; 2Department of Chemical Sciences, University of Johannesburg, Doornfontein 2028, South Africa

**Keywords:** cellulose nanoparticles, biodegradable polymers, nanocomposites, foaming, physical properties

## Abstract

The interest in designing new environmentally friendly materials has led to the development of biodegradable foams as a potential substitute to most currently used fossil fuel–derived polymer foams. Despite the possibility of developing biodegradable and environmentally friendly polymer foams, the challenge of foaming biopolymers still persists as they have very low melt strength and viscosity as well as low crystallisation kinetics. Studies have shown that the incorporation of cellulose nanostructure (CN) particles into biopolymers can enhance the foamability of these materials. In addition, the final properties and performance of the foamed products can be improved with the addition of these nanoparticles. They not only aid in foamability but also act as nucleating agents by controlling the morphological properties of the foamed material. Here, we provide a critical and accessible overview of the influence of CN particles on the properties of biodegradable foams; in particular, their rheological, thermal, mechanical, and flammability and thermal insulating properties and biodegradability.

## 1. Introduction

Biodegradable polymers have gained enormous attention in both academia and industry over the past decades and are still of immense interest. The interest in this kind of material stems from the negative concerns regarding most fossil fuel–derived non-biodegradable polymers. The use of petroleum-derived polymers has lasted to many decades. Their broad utilisation in many fields of application emerges from their availability at low prices and the properties they offer, such as good mechanical and thermal properties, among others [1,2,3,4]. In addition, they can be recycled, although it is not possible to recover all the plastics used, and recycling is an energy-consuming process [5,6]. However, certain properties, including rapid biodegradation, cannot be achieved with these materials. The non-biodegradability and persistence in the natural environment and the consequent pollution are the major concerns regarding these materials. For instance, the agricultural sector generates approximately 10% of the plastic waste in landfills, while packaging applications are believed to generate more, as most of the plastic materials are used in this sector [7]. Consequently, there is a need to develop materials that are biodegradable in the natural environment as a substitute to non-environmentally friendly materials. Further, this is motivated by the projections regarding the rapid increase in the human population, which is expected to reach 8.6 billion in 2030 and 9.8 billion in 2050 [8]. Bearing this in mind and considering the contribution of the human population toward pollution due to various practices such as illegal dumping and incineration, it is necessary to develop greener materials that are compatible with the environment. Hence, the development of biodegradable polymer materials is crucial and is desired for various kinds of applications.

Polymers are used for a variety of applications, as depicted in Figure 1 [9]. It is evident from the figure that polymers are mostly used in plastic packaging. In packaging applications, plastic food packaging is the principal sector where polymers are employed [3]. Plastic foams have also found a wide range of applications, including packaging, cushioning, shock absorption, and insulation, because of their low weight. In some cases, they also offer better properties than unfoamed plastics. The most commonly used polymer foams are polystyrene (PS) and polyurethanes (PU) foams. Others include polyolefins, such as polyethylene (PE) and polypropylene (PP) [10,11,12]. However, these foams are petroleum oil–based materials and are not biodegradable; further, their collection and storage for recycling are inconvenient.

The feasible strategy to minimise the utilisation of non-biodegradable polymer foams is to develop bio-based polymeric foams as a substitute for these materials. Because of their compatibility, biodegradability, and renewability, biodegradable polymer-based foams can be employed for a broad spectrum of applications where biodegradation is a key. Figure 2 shows the publication history (a) and publication statistics (b) on biodegradable polymers from 2008 to 2018. It can be clearly noticed from the trend in Figure 2b that in general, research on biopolymer foams has not received the expected attention, as compared to that on biodegradable polymers. However, it is expected that there will be an increase in the number of publications on biodegradable foams in the coming decade, which will pave the way for scientific research to enable the mass production of biodegradable polymer foams.

Biodegradable polymers such as poly (lactic acid) (PLA) and others are considered as alternatives to petroleum-based foams [13]. However, the melt strength, melt viscosity, and crystallisation rates, which are fundamental prerequisites for polymer foaming, are low for biodegradable polymers, including PLA and poly (hydroxybutyrate-hydroxyvalerate) (PHBV). Low melt strength results in cell coalescence and cell rupture during cell growth. To mitigate this shortcoming, enhanced crystallisation rates are required to compensate for the low melt strength and melt viscosity [13,14,15] and can be achieved by blending with other polymers that crystallise much faster. The incorporation of nanofiller materials in biopolymers is another alternative to improve the low melt strength and viscosity and, thus, enhance foaming [1,14,15,16,17]. Nanofiller materials derived from biomass, such as cellulose nanostructure (CN) materials, are suitable for the development of biopolymer-based nanocomposites and foams because of their low density, high aspect ratio, large surface area, non-toxicity, and, most importantly, biodegradability and environmental friendliness [18,19,20]. During foaming, CN particles act as nucleating agents to enhance the formation of nucleation sites and, thus, control the morphology of the foams. In addition, they contribute to other foam properties, such as mechanical and thermal stability, due to enhanced crystallinity of the material. In the absence of nucleating agents, the foam cells become overly large for a small number of cells, thereby leading to low cell density [21,22]. Hence, CN particles can be used to improve foam properties. Studies have shown the effectiveness of these materials in reinforcing polymer materials [23,24,25,26,27,28]. In this review, we critically discuss the influence of CN particles on the properties of biodegradable foams; in particular, their rheological, thermal, flammability and thermal insulation, and mechanical properties. Furthermore, the concepts reviewed here, among others, include the influence of processing parameters on CN-reinforced nanocomposite foams and the processing techniques used to develop CN-based biopolymer foams.

## 2. Biodegradable Polymers and Their Classification

Biodegradable polymers are polymers that can be broken down and catabolised into both carbon dioxide and water by microorganisms in the natural environment. Such degradation usually occurs via a two-step process. The first step involves the breakdown of the polymer into lower-molecular-mass species by means of abiotic or biotic reactions. The second step is bio-assimilation of the polymer fragments by microorganisms and their mineralisation [3,6,29]. Various aspects can be considered to classify biodegradable polymers. Table 1 shows the classification of biodegradable polymers based on their origins, families, and common examples in each category. A detailed classification and discussion of biodegradable polymers are provided elsewhere [2,3,6,29,30,31,32,33]. Nevertheless, among the different biomass-based polymers, cellulose has outstanding characteristics that make it favourable, whereas starch has difficulties in processing and is brittle [34]. In addition, cellulose is abundant, as it can be obtained from various sources ranging from plants to microorganisms.

## 3. Cellulose

The most abundant natural polymer found on Earth is cellulose. Discovered by Anselme Payen back in 1838, this renewable and biodegradable material has received enormous attention because of its physical and chemical properties, which are different from those of synthetic polymers. Cellulose can be obtained from various sources, such as plants, bacteria, and algae. The cellulose obtained from bacteria is purer than plant-derived cellulose, as the latter also contains hemicellulose [34,35,36,37,38]. Figure 3 shows the structure of cellulose. Cellulose was one of the key research areas during the emergence of polymer chemistry, and its structure was first demonstrated by the pioneer in polymer science, Herman Staudinger, in 1920 [34,37]. An in-depth understanding of this material is crucial for fine-tuning of its parameters for various purposes. This requires a detailed understanding of its structure, reactivity, and possible modification routes that can be followed.

### 3.1. Cellulose Nanostructures

CNs is cellulosic materials with dimensions in the nanometre range. They can be obtained in two forms: as cellulose nanofibers (CNFs), also termed microfibrillated cellulose (MFC), and cellulose nanocrystals (CNCs), which are also called cellulose nanowhiskers (CNWs). The former is obtained by the mechanical processing of cellulose in a high-pressure homogeniser into smaller particles, while the latter is obtained by deterioration of the amorphous phase of cellulose by hydrolysis with inorganic acids to attain nearly completely crystalline cellulose. CNFs, which are long and flexible, have a diameter in the range of 5–60 nm, whereas CNCs vary in length from 100 to 300 nm and width from 5–70 nm. However, these dimensions differ depending on the source and preparation method. As depicted in Figure 3, CNs consist of repeating D-glucopyranose units linked via equatorial groups of carbon (C) 4 and C1 glycosidic linkages (β-1,4-glycosidic linkages). Every repeating unit is corkscrewed 180° around its backbone chain with respect to its neighbours. The hydroxyl (–OH) groups on the repeating units form strong hydrogen bond networks between the molecular chains. Thus, the cellulose chains align straight and assemble into crystallite strands [34,35,39,40].

Both CNCs and CNFs are hydrophilic in nature because of (–OH) groups on the backbone, which poses a challenge when incorporating them into hydrophobic polymer matrices. Therefore, they require modification prior to application. Such modification can be achieved by utilising the (–OH) groups on their backbone to form covalent bonds with other substances, which makes CNs partially hydrophobic. However, several factors may affect the reactivity of these groups, including steric effects, which result from the reacting agent and supramolecular structure. The reactivity of (–OH) groups on the cellulose backbone is as follows: C6-OH > C2–OH > C3-OH (Figure 3) [34,41]. Jain et al. [42] found that the reaction rate for the primary (–OH) (C6–OH) of cellulose is 43 times higher than those of secondary (-OH) groups towards chlorotri (*p*-tolyl) methane.

### 3.2. Modification of Cellulose Nanostructure

Several studies have dwelt on the modification of CNs, their use as a reinforcing agent in nanocomposites, and their processing methods [23,27,36,41,43,44,45,46,47]. Moreover, several methods for modifying CNCs and CNFs have also been discussed in these reports, including modification by a reaction with low-molecular substances via acetylation, silanisation, silylation, and other reactions and via grafting of polymer chains or oligomers from/to or through these groups. Surface acetylation of CNCs with acetic anhydride was carried out by Lin et al. [23]; the obtained CNCs were then used as a reinforcing agent in PLA. The acetylation of CNCs was confirmed by Fourier-transform infrared (FTIR) spectra, which showed a carbonyl group (C=O) peak attributed to the ester group formed on the CNC surface. Furthermore, contact angle measurements were carried out to examine the polarity of the acetylated CNCs (ACNCs). The results indicated an increase in the contact angle of the polar medium (water) from 44.7° to 78.0°, suggesting a high degree of ACNC hydrophobicity, and a decrease in the contact angle of the non-polar solvent (diiodomethane) from 19° to 12.1°, indicating high affinity of the ACNC to the non-polar solvent. Silanisation treatment of CNFs was performed using methyltrimethoxysilane (MTMS). This was also confirmed by FTIR spectra, which showed vibrations for Si-OH bonding in silane groups and Si-C vibrations. Contact angle measurements (133.51° ± 1.89°) indicated a decrease in the affinity of the modified CNFs to polar solvents, implying that the CNFs became hydrophobic [43]. A hydrophobic material should have a contact angle greater than 90°; therefore, the silanisation modification was successful.

Grafting of polymer chains or oligomers from the cellulose backbone is another possible method of modifying the surface of CNs. The ring-opening polymerisation of cyclic monomers is normally initiated by (–OH) on the surface of cellulose. Therefore, cellulose modification prior to grafting is not necessarily required in this case. In one study, CNC was grafted with butylene glutarate (BG) oligomers via the ring-opening of glutaric anhydride (GA) [44]. GA first reacted with a CNC suspension in toluene for 4 h, following which 1,4-butanediol was added to form short chains of butylene glutarate (PG). To observe the effect of grafting on the hydrophilicity of the CNC, water absorption studies were conducted, and the results showed a decrease in the saturation water content to 1.9% compared with the unmodified CNC, which had a saturation water content of 2.8%. Nuclear magnetic resonance (NMR) also confirmed this modification. Lönneberg et al. [45] also grafted CNFs with poly (ε-caprolactone) (PCL), which was shown to be dispersible in a non-polar solvent (tetrahydrofuran; THF), indicating the hydrophobisation of CNFs. FTIR spectrum further corroborated the functionalisation of CNFs by showing the C=O group peak in the PCL structure. Peltzer et al. [48] developed a grafted copolymer from CNCs and L-lactide monomer via the ring-opening of the cyclic l-lactide monomer. Successful modification was confirmed by the long-term stability of the modified CNCs in chloroform, which is a non-polar solvent, and by FTIR and X-ray photoelectron spectroscopy (XPS) analyses.

### 3.3. Preparation Methods for Cellulose Nanostructure-Based Nanocomposites

CN materials are used as reinforcing agents in polymers because of their high aspect ratio. Prior to foaming, CN-based nanocomposite materials are prepared and then characterised to evaluate the dispersion, distribution, and compatibility of CNs with a polymer. Additionally, the effect of these nanostructured materials on the matrix properties, such as rheological and crystallisation behaviour, is evaluated. Generally, two methods are mainly used for the development of cellulose-based nanocomposites: the ex situ method, which involves the melt-compounding and mixing of polymer solutions with a suspension of cellulose, and the in situ method, in which polymer chains are grown from the cellulose backbone [36]. However, these methods have their respective drawbacks, which restrict their application. For instance, an ex situ method like solution casting presents a difficulty in scaling to the industrial level and lacks homogeneity in the filler dispersion because of poor bonding between the organic matrix and filler. Melt-mixing is another ex situ method and can be readily employed at the industrial scale; however, the dispersion and interfacial bonding of cellulose with the polymer matrix may not be improved to the desired level via this method. In this method, the nanocomposites are prepared in internal batch mixers or extruders [36,41]. In situ (grafting from cellulose backbone) polymerisation of polymers from the surface of cellulose emerged as a preferred method because of the uniform dispersion of cellulose. The strong interfacial adhesion between the cellulose and polymer matrices can be achieved with the growth of polymer molecules, via this method. Moreover, chain entanglements can be obtained, which improves mechanical properties; co-crystallisation may also occur, thereby enhancing interfacial adhesion between the matrix and fibre [36,45]. Table 2 summarises the various CN-based biodegradable nanocomposites prepared by ex situ and in situ methods.

Among the processing methods for CN-based nanocomposites, extrusion seems to be the most efficient technique for large-scale production. However, interfacial adhesion between the CN particles and polymers remains a challenge with this process. The alternative is to modify the CN particles separately and then incorporate them into the matrices via extrusion. This step is important for improving the interaction between the filler and matrix. Thus, enhanced properties can be achieved because of the improved interaction by CN modification.

## 4. Role of CN in Foam Processing

During foaming, CN nanoparticles act as nucleating agents. The term “nucleation” in foam processing refers to the formation of a new phase (bubbles/cells) through a reversible thermodynamic process within a continuous polymer phase. This step is followed by the growth and stabilisation of the nucleated bubbles. According to the classical nucleation theory (CNT), a bubble should have a radius (R) equal to or greater than the critical radius (r^*^) before it can become stable. At the critical radius, the free energy is maximum (Figure 4). Below the critical radius, the nucleated bubble collapses, while it survives above this radius [68,69,70]. Nucleation can occur in three forms: homogeneous nucleation (when an amount of gas is dissolved in the polymeric phase to form stable bubbles), heterogeneous nucleation (at the interface of the polymer and additive), and mixed-mode nucleation (from the competition between homo- and heterogeneous nucleations in the transition regime) [71].

Heterogeneous nucleation occurs in the presence of nucleating agents, which act as nucleation centres. In this regard, CN particles produce an increased number of nucleation sites because of their large surface-to-volume ratio and strong interaction with a polymer. Therefore, CN particles are expected to yield higher cell density with a smaller cell size. Such behaviour has been reported in the literature and will be discussed in the following sections. In Figure 4, it can be observed that during heterogeneous nucleation, the presence of nucleating agent results in a decrease in the activation energy for bubble nucleation, when compared with homogeneous nucleation, which initiates at high energies. Bubbles are formed either at low temperatures or decreased pressure. However, the reduction of free energy is controlled by the size, shape, and topography of the nucleating particle [68,72,73,74]. Moreover, the size of the cells decreases in the presence of nucleating agents because of the increased melt viscosity and strength, which reduce bubble expansion. In addition, the melt viscosity increases in the presence of nucleating agents and, as a result, the induced strain hardening also leads to a decreased cell size [1,10]. According to the CNT, heterogeneous nucleation occurs at the interface of the nucleating agent and continuous polymer melt phase mainly because of the high thermodynamic instability (increased temperature and decreased pressure when bubbles are formed) occurring at the interface [68]. Models for bubble nucleation have been discussed in the literature [71,75,76]. For heterogeneous nucleation, the Gibbs free energy can be estimated from Equation (1) [71].
ΔG* _het_ = 16 π γ_bp_^3^ (3 ΔP^2^)^−1^ S (θ)(1)

Here, γ_bp_ is the bubble–polymer interfacial tension, ΔP is the pressure of the gas in the bubble, and S is the shape factor, which depends on the wetting angle (θ). 

As for BAs, nucleating agents should possess certain properties for optimum performance when applied in foaming. These properties are as follows [68]:There should not be strong adhesion to the polymer matrix;Nucleating agents should disperse uniformly and be exfoliated. However, intercalation may also be desired for increased thermodynamic fluctuations at the matrix-filler interface;Their amount should be adequate to increase the number of nucleation sites. Excess nucleating particles may lead to agglomeration.

## 5. Methods and Parameters Associated with Cellulose Nanostructured Nanocomposite Foam Processing

Various processing methods are used for the fabrication of biopolymer foams, and they are summarised in Table 3. Three types of processing techniques that are primarily used for the development of CN-biodegradable foams include batch processing, extrusion foaming, and injection foaming [13,68]. The aforementioned three methods follow the same foaming principles. First, the polymer is saturated with the BA. Then, the thermodynamic stability necessary for cell nucleation is induced by either decreasing the pressure or increasing the temperature. Finally, the growth and stabilisation of the cells is achieved by gas expansion from the polymer into the nucleated bubbles. Among the three methods, batch processing is the least expensive because of its scale; moreover, it is non-continuous. Okolieocha et al. [68] have thoroughly reviewed foaming methods. Among the factors used to distinguish these processes, one can identify the governing factors for the formation of cellular structures.

In batch foaming, the driving factors for cell nucleation are pressure drop and temperature increase, while in extrusion and injection foaming, it is depressurisation. Furthermore, a uniform cell size distribution can be attained in both batch processing and extrusion, although the radius might differ at the edges and in the middle of the foam during extrusion. In foam injection moulding, however, a uniform distribution is not easily attained because of the coupling of filling and holding times, which results in nucleation at different locations.

In addition to these techniques, other foam processing methods that do not require BAs have been used. These include thermally induced phase separation (TIPS), phase inversion, and casting and leaching methods. All these methods are solvent-based processes. In the casting and leaching method, the polymer has to be dissolved in a volatile solvent and then casted into a porogen-containing medium. The porogens are pore-forming materials, which can be leached out at the final stage of processing. Typical examples include solid particles, such as salts and sugars. However, porogens can also be in the form of liquid or gas. Furthermore, the size of the porogen material dictates the size of the pores. Therefore, the pore size can be altered by simply changing the porogen size [10,95,96]. Borkotoky et al. [97] fabricated PLA/CNC foams using the casting and leaching method, using sucrose particles as porogens. Both PLA and CNC (at 1%, 2%, and 3%) particles were mixed together in 1,4-dioxane solvent at 70 °C followed by addition to the sucrose solution which was also dispersed in the same solvent and at the same temperature for 12 h. Water was used to leach out the sucrose porogen particles. Some of the obtained foam properties are summarised in the following sections.

### 5.1. Batch Processing

Batch foaming is normally carried out in an autoclave system. Foaming of the material via this process can occur in two different ways—namely, the pressure-quench method and temperature-induced method. In the former method, the polymer is saturated with gas until the equilibrium is reached, and, as a result of rapid pressure drop, cell nucleation and growth occur. The final step here is cooling of the foamed product in either a solvent or air. Temperature-induced batch processing is similar to pressure-induced batch processing. In this case, the saturated polymer material is taken out of the autoclave and placed in a hot solvent or oil bath, where cell nucleation occurs. The final step is also similar here—cooling by immersing the sample in solvents or water [68]. CN/biopolymer foams reported in the literature have mostly been fabricated by batch processing. Dlouhá et al. [98] prepared PLA foams reinforced with CNF using the pressure-quench batch method at 60 °C. The samples were conditioned for 6 h in supercritical CO_2_. In their work, the authors varied the foaming pressure from 12 to 20 MPa with the depressurisation time less than 3 s. Nucleation was induced by a rapid pressure quench to atmospheric pressure. It was observed that the increase in foaming pressure from 12 to 14 MPa resulted in changes in the cell morphology of the foams. A decrease in cell density was observed for neat PLA. However, above 14 MPa, the cell size of the nanocomposites did not evolve significantly, while for neat PLA, the cell size decreased further with pressure. Similar behaviour was observed for the cell densities of the tested foams. The cell density initially increased with a pressure increase up to 14 MPa and increased further in the case of neat PLA. For nanocomposite foams, the cell density did not significantly change with pressure. Qiu et al. [99] prepared PLA/CNC foams by temperature-induced batch foaming. Initially, the prepared nanocomposites were placed in a high-pressure vessel and then saturated with CO_2_ at a pressure of 5 MPa and a temperature of 0 °C for 12 h. The CO_2_ pressure was quickly released from 5 to 0.1 MPa within 3 min. Nucleation was initiated by immersing the saturated polymer composite in a water bath at 60 °C for 30 s.

In a different study [100], compression-moulded PBS/CNC nanocomposite foams were prepared using a hot-press. Prior to compression moulding, PBS/CNC was melt-compounded with azodicarbonamide (AC) (CBA) and zinc oxide (ZnO) (blowing promoter) in an internal mixer at 120 °C with a rotating speed of 72 rpm for 15 min. For foaming, the temperature of the hot-press was set to 167 °C to allow decomposition of the BA. The AC concentration was initially varied in the range of 3–6 wt.% to determine its optimum value. At this stage, CNC (3 wt.%) was added into the PBS/AC system to avoid the destruction of the microstructure and interestingly, the PBS/AC foams with CNC particles exhibited evenly distributed cells with small size and more stable compared to neat PBS/AC without CNC. The type of developed morphology was shown to be controlled by the AC content. For 4 and 5 wt.% AC, a spherical morphology was observed for even and well-distributed cells, while an oval-shaped morphology was obtained at 6 wt.%. Therefore, an AC concentration of 5 wt.% was selected for the foaming of PBS/CNC nanocomposites.

### 5.2. Injection Foaming

Injection foaming is a semi-continuous foam process used to produce shaped foam parts. Foam injection moulding was invented in the 1950s, during which period a small pinch of baking powder was added to remove sink marks appearing on the product. Although this had advantages, including reduced processing time, eliminated sink marks, warping, and lower material costs, the technique still has drawbacks such as non-uniform cell morphology, which results from nucleation at different spots, and low density in the range of 10^4^–10^8^ cells/cm^3^ compared with batch and extrusion foaming. Injection foaming is similar to conventional injection moulding. However, injection foaming requires an injection gas unit where the gas is introduced into the polymer melt. Injection foaming can be classified into two types: the low-pressure process and high-pressure process. Supercritical nitrogen gas is normally used in injection foaming because of its stronger nucleation force [13,68,101].

In commercial applications, there are three types of injection foaming units that are widely used: (i) MuCell^®^ by Trexel Inc.; (ii) Optifoam developed by Institute of Plastic Processing (IKV) Sulzer Chemtech; and (iii) Ergocell developed in 2001 by Demag [102]. All studies used super-critical fluids (SCF). Fifteen years later, Ergocell, Yusa, et al. [103] performed foam injection moulding using no supercritical fluids. PBAs via this type of injection moulding are introduced into the barrel without pressurising them into a supercritical state. However, they are directly transferred from their cylinders into the molten polymer through injector valves. With the specific design of the screw configurations, the PBAs can be transferred directly into the polymer melt. The advantage of this technique is that the excess gas can be removed from the machine through vent holes. Likewise, when the level of saturation is achieved, the PBA can be delivered back to the molten polymer through the venting holes. With this technique, the authors managed to prepare microcellular propylene (PP) with a cell diameter of less than 25 μm and cell density greater than 6.0 × 10^7^ cells/cm^-3^.

PCL/CNC foams have also been prepared by microcellular injection moulding (MuCell Trexel, Inc.) using scCO_2_ (controlled at 1.5 wt.% of PCL), which enabled control of the weight of the injected gas [104]. Scanning electron microscopy (SEM) images indicated that the pore sizes were not uniform (Figure 5), which was attributed to the processing of the materials, which may have involved non-uniform cooling of the foamed sample, complex mould design, and shear flow caused by injecting the molten material into the mould cavity from a small gate. These are common occurrences during this type of processing.

The SEM images in Figure 5 indicate that injection processing has a great effect on morphology development. Although the addition of CNCs resulted in improvement in the cell density and cell size, the processing parameters need to be optimised in order to obtain microcellular foams with uniform cell size and higher cell densities.

### 5.3. Extrusion Foaming

Extrusion foaming is a continuous process and is industrially used for foaming on a large scale [13,68,105]. This technique is similar to plastic extrusion which is normally used to produce plastic items, as it consists of a feeding zone, mixing zone, and die zone. However, in extrusion foaming, an injection unit is added to the barrel to introduce gas into the molten polymer. The second major difference is that the extrusion foaming equipment has two extruders: a melt extruder and a cooling extruder. Similar to normal extrusion, the polymer is first introduced into the melt extruder through the hopper. As the polymer melts and mixes, the gas is introduced from the syringe pump into the melt to form a polymer-gas solution under high pressure. The second phase involves the transfer of the polymer–gas solution to the second extruder, where the mixture is initially cooled to temperatures below the melt extrusion temperatures. However, at this stage, cell nucleation does not occur, due to the high pressure developed inside the barrel. At the final stage, the mixture is advanced into the die zone. As the material exits the die, cell nucleation is induced by rapid pressure drop, and the cells start to grow until vitrification occurs [13,68,105,106]. Several factors affect the cell density and cell morphology during foam extrusion. The die geometry and operating temperatures influence the cell density and cell morphology [72,105]. The influence of die temperature on the morphology of poly (vinyl alcohol) PVOH/CNF (0.05 wt.%) prepared with 9 wt.% sc-CO_2_ and 12.5 wt.% water as a co-BA was investigated [107]. The die temperature was varied from 140 to 180 °C, and the results are shown in Figure 6.

At a very low die temperature (140 °C), the cell size was small, while the cell density was the maximum. This was because a decrease in the die temperature results in improved melt strength and viscosity of the material and, thus, reduced cell coalescence, which thereby enhances the cell size as well as density. In addition, the authors showed that an increase in the die pressure at low die temperatures can ensure complete dissolution of the gas into the polymer, which enhances the thermodynamic instability at the die exit and, consequently, enhances cell nucleation at low temperatures. Chauvet et al. [108] also observed an increase in the cell density with a decrease in the die temperature on PLA starch–based foams. Based on the obtained results, the authors preferred the foaming of PVOH to be performed at a die temperature below 160 °C because of the obtained small cell size and large cell density.

In all these methods, the processing conditions were critical for obtaining cellular structures with low cell diameters and high cell density. To obtain the desired foam morphologies, nanocomposite foam processing requires process optimisation to attain specific processing conditions for a certain type of polymer nanocomposite material. From the different processing techniques and conditions used to develop cellulose-based biodegradable nanocomposite foams, it was clearly observed that the type of cell morphology solely depends on the processing methods and conditions. In batch foaming, it has been shown that the foam morphology is dependent on the foaming pressure [98,99]. The optimisation of pressure in this case is important to achieve foam with high cell density and small diameters. In addition, the concentration of the CBA also determines the type of morphology that develops in the foams, as indicated by Lin et al. [100]. The processing temperatures, particularly the die temperature, and the pressure have a significant influence on the cell properties in foam extrusion, as shown in a previous study [107]. In injection foaming, several factors affect the foam morphology that develops in the polymer foam. Mi et al. [104] observed the non-uniform cells and attributed this to the cooling time and design of the mould. In this case, the morphology of the foamed item was not uniform throughout the item, and the overall performance of the foamed part might differ because of the difference in the sizes of the cells across the item. Consequently, the processing–morphology–property relationship comes into play, as the final behaviour of the foamed part would strongly depend on the type of the obtained cell structure. Therefore, optimisation of the processing conditions in each technique is very important to obtain the desired morphological properties and better performance of the final part. However, this requires an understanding of the foamed material, especially when fillers are incorporated. Therefore, preparation and characterisation of the material is required prior to foaming with any of the techniques.

## 6. Influence of Cellulose Nanostructures on Crystallisation and Morphological Properties of Foam

Crystallinity is one of the controlling factors in the foaming of a polymer material. The cell morphology and expansion ratio of the material are affected by the presence of crystallites in a polymer. Crystallite-induced heterogeneous nucleation has been shown to increase the cell density of foams and decrease the cell size. However, not only the extent of crystallinity, but also the crystallite size affects the foam morphology. A smaller crystalline size and higher crystal density can induce heterogeneous nucleation effectively and, thus, lead to the formation of uniform cells with a smaller size and higher cell density. On the other hand, large crystallites tend to inhibit the formation of uniform cells throughout the foam [109,110]. However, in the presence of nanoparticles, the crystal size decreases and the crystal density tends to increase. Ji et al. [109] observed a decrease in the PLA crystalline size upon the addition of nanosilica particles in the presence of CO_2_. Furthermore, in multiphase systems, such as blends and nanocomposites, the nucleating efficiency of the filler material can be observed either by the increased crystallinity or higher crystallisation temperatures. Enhanced crystallisation temperatures suggest that the melt solidifies quicker and, thus, avoids overgrowth of the cells. In addition, in the presence of nanoparticles, more bubbles are nucleated, which results in a reduced cell size due to the lesser amount of gas available for cell growth. Further, an increase in the crystallinity of the material contributes to the improvement of other foam properties, such as mechanical properties. Therefore, CN particles show a synergistic effect in improving both cell nucleation and crystallinity of the material and, consequently, improve other foam properties. In this section, we study the influence of CN particles on the crystallisation and foam morphology of various biodegradable polymers.

CN nanoparticles in nanocomposite materials have proven to be effective in enhancing the crystallisation and foaming behaviour of various biodegradable polymers. According to the CNT, heterogeneous nucleation occurs due to bubble nucleation at the polymer-particle interface. Therefore, it is expected that by increasing the number of nucleation sites via adding CN materials, the cell density would increase because of nucleation at many locations. In other words, the rate of heterogeneous nucleation depends on the number of nucleation sites available for cell nucleation, which in turn depends on the concentration of particles in a composite [10,68]. Zhao et al. [107] examined the foaming behaviour of (PVOH)/MFC. The effect of MFC on enhancing the crystallisation of PVOH was observed by loading 0.25 and 0.5 wt.% MFC, by which the nucleating efficiency of MFC was confirmed. At 0.5 wt.%, the crystallinity and crystallisation temperature (T_c_) increased by 2.2% and 4 °C, respectively. Although T_c_ increased after MFC addition, one can say that the improved T_c_ did not contribute to foaming. Specifically, in this case, ignoring the influence of MFC concentration on T_c_, it was observed (Figure 8) that foaming occurred above the T_c_ of PVOH. This clearly shows that by nucleation, foaming can effectively occur in a melt state because at that stage, the melt viscosity and strength are sufficiently high to sustain cell nucleation and growth at the polymer-filler interface even prior to the formation of crystals. A high T_c_ contributes significantly toward enhancing the solidification of the material in order to avoid overgrowth of the cells. However, at this stage, cell nucleation occurs around the nucleated crystals, thereby contributing to the cell density. A uniform cell structure with small cell size and increased cell density was obtained with MFC loading up to 0.05 wt.%, which was the optimal content for maximum cell density (1.25 × 10 ^10^ cells/cm^3^). This was associated with the increased local pressure variations due to the nucleating efficiency of MFC in enhancing the cell nucleation rate. Above this loading, a decrease in cell density and an increase in cell size was observed, which was attributed to overloading of the MFC relative to its dispersing ability. Although the CNF was efficient in enhancing the PVOH foamability, it was not effective in promoting the foamability of PHBV [111]. In that study, although the crystallinity of the material was high in the presence of CNF, the foaming of PHBV was hindered, possibly because of the decreased solubility of CO_2_ in the crystalline regions. Another possible reason is that the melt strength of the material was too high in the presence of CNF to allow cell nucleation.

PBS/CNC foams using AC as a BA were investigated [100]. With respect to neat PBS/AC foams, the crystallinity degree of the PBS/CNC foams at 5% AC was found to increase by 2.9 and 6.3% at 3 and 5% CNC concentrations, respectively. The cell morphologies of the foamed nanocomposites with varying CNC concentrations at a fixed amount of AC (5%) are depicted in Figure 7. It can be seen from the SEM images in Figure 7 that at 5 wt.% CNC, smaller homogeneously distributed cells are obtained in comparison with 3 and 10 wt.% CNC. At these concentrations, especially at 10 wt.% CNC, the non-uniform distribution and larger cell size could result from the aggregates that formed and microphase separation at this level. Figure 8 shows that the increase in the CNC content resulted in the increase in cell density, possibly because of the increased number of nucleation sites, while the cell size decreased with increase in the CNC content. At 5 wt.%, the maximum obtained cell density was 7.1 × 10^5^ cell/cm^3^, which was a 69.1% increase with respect to the neat PBS foam. The average cell size was 187 μm. Thus, 5 wt.% was the optimum amount to reach the maximum cell density.

Similar behaviour was observed in PCL/CNC foams, as shown in Figure 9 [104]. The maximum cell density was obtained at 5 wt.% CNC and was approximately 833% higher than that of neat PCL foam. At this concentration, the degree of crystallinity with respect to the neat PCL matrix increased from 39.0% to 41.4%, while T_c_ significantly increased from 34.2 °C to 36.7 °C. These results indicate that CNC contents up to 5% are adequate to obtain the maximum cell density in PCL and PBS foams.

Borkotoky et al. [97] observed changes in the crystallinity of PLA upon the addition of CNC. The XRD results showed the α-crystal to be dominant in PLA foams, indicating the nucleating efficiency of CNC in improving the crystallisation of PLA. The calculated crystallinity improved with an increase in the CNF content from 42.9% neat PLA to 43.6, 44.0, and 55.6% with 1, 2, and 3% CNC concentrations, respectively. An open-cell structure was obtained in all investigated foams. Because of the higher number of nucleation sites generated by CNC nanoparticles, the number of pores and, thus, the cell density increased, while the pore size decreased in comparison with neat PLA foam. The obtained results (cell size and density) are summarised in Table 4. As can be seen in Table 4, a small cell size and large cell density were obtained at 1% CNC. This might be due to better dispersion at that CNC concentration.

Further addition of CNC led to an increase in the cell size and decrease in the cell density, although they were still improved compared with neat PLA. In all these investigations [97,100,104,107], the cell morphology was shown to depend on the CN concentration irrespective of the processing method used. With an increase in the number of CN particles, the number of nucleation sites increased, thereby improving the cell density. In addition, the cell size decreased with the increase of CN content, which could be caused by the enhanced melt strength and viscosity, which restricted overgrowth of the cells. Borkotoky et al. [97] found improvements with only 1 wt.% CNF particles, while above this, the cell properties were deteriorated. Lin et al. [100] observed a decline in cell properties above 5 wt.% CNC. The observed decline can be due to poor interaction between the particles and the matrix, as a result of CN agglomeration above certain concentrations. Similar behaviour has also been reported, when the maximum cell density and minimum cell size were obtained at 5 wt.% CNC in PCL-based foams [104].

Qiu et al. [104] prepared PLA foams containing both acetylated-modified CNC and pristine CNC. The PLA and its nanocomposites in this study were completely amorphous, due to quenching from the melt. Hence, the crystallisation behaviour was completely neglected. However, the presence of CNC was effective in enhancing the morphology of the prepared foams, which was associated with the nucleation effect of CNC particles and reduced cell coalescence of the bubbles due to stabilized nucleated cells and, possibly, a reduced energy barrier of nucleation. A decrease in the energy barrier of nucleation prepones and accelerates nucleation. Furthermore, the acetylation of CNC, as compared to native CNC, significantly enhanced the foam morphology than the pristine. This was related to the high degree of dispersion of the modified CNC, which improved the interaction with PLA. Nonetheless, well-dispersed nanoparticles can improve cell nucleation. Therefore, distributed particles can be favourable because they are more rigid and increase the local pressure variations necessary for cell nucleation.

Thus, CN nanoparticles are effective nucleating agents for foaming and improve the crystallisation of various polymers. Their large aspect ratio, which results in increased filler-matrix interaction sites, increases the cell density because of the higher number of nucleated cells. Furthermore, the cell size decreases because of the restriction of cell overgrowth and enhanced melt strength by incorporation of the nanoparticles. However, at higher concentrations, the CN nanoparticles aggregate, resulting in non-uniform cells with different sizes. Therefore, an optimum amount is required to achieve maximum cell density and cell size.

## 7. Properties of Cellulose-Nanostructured Nanocomposite Foams

### 7.1. Rheological Properties

The viscoelastic properties exhibited by polymer materials are important in determining their foaming behaviour. In particular, the melt strength and melt viscosity are the most important properties because they control the cell morphology in stabilising the nucleated bubbles during processing. Biodegradable polymers, including PLA and PHBV, have low melt strength and viscosity, which inhibits their foaming [13,14]. Poor melt strength and viscosity result in cell clustering and cell rupture. At low melt strength, cell formation and growth occur easily; however, nucleated bubbles overgrow into larger cells, which consequently collapse. On the contrary, high melt strength leads to insufficient growth of the cells and inhibits the formation of the cells. Therefore, moderate melt strength and viscosity are necessary to stabilise and control the size of the nucleated bubbles [112]. The melt strength and melt viscosity can be evaluated by determining the storage modulus and complex viscosity, respectively [113]. Several studies have indicated that the incorporation of filler materials or blending with other polymers can enhance these parameters and, thus, the foam morphology. For instance, Liu et al. [114] enhanced the melt strength of benzoyl peroxide-crosslinked-PLA by blending it with PBAT. Zhou et al. [115] also observed an increase in both, the melt strength and viscosity, of PBS/PLA reinforced with organically modified montmorillonite (OMMT). Consequently, the morphologies of the prepared foams were significantly affected. Studies have also revealed that enhanced crystallisation rates can also compensate for poor rheological properties of biodegradable polymers [13,87,116]. In the presence of a crystal nucleating agent, high melt strength can be achieved because of the induced crystals, which connect the polymer molecules into a long-chain branched material [107].

The incorporation of nanoparticles increases the melt viscosity of the nanocomposites, which results in the reduced rate of cell expansion. The CN materials used as nanofillers have also been proven to be effective in enhancing both melt strength and viscosity of biopolymers and their foamability. Cho et al. [117] observed the enhancement of foaming properties of PLA foams filled with acetylated CNF as a nucleating agent. A decrease in cell size from 52.6 (neat PLA) to 37.3 (1% CNF), 32.9 (2% CNF), and 26.7 (5% CNF) μm was explained through the rheological properties of foamed composites, which showed the increase in complex viscosity due to immobilised PLA chains by CNF particles, which consequently obstructed the growth of the cells. The cell density also increased with increasing CNF concentration, probably because of the increased number of nucleation sites with the addition of the filler. As mentioned above, PLA exhibits poor rheological properties because of its linear structure and high rigidity, which inhibit rapid chain entanglement necessary for foaming; therefore, cells are expected to merge and collapse, which would, in turn, affect foam properties. However, the addition of modified CNF at 1 and 2 wt.% resulted in the improvement of melt viscosity, which resulted in uniform cell morphology with reduced cell size. According to literature, rheological properties increase with the increase of filler concentration; it was also observed in study done by Cho et al. [117].

Nevertheless, at 5 wt.%, the decrease of complex viscosity even at low shear rate was observed, which was referred to the aggregation of CNF at high amounts. Despite a decrease of viscosity at this concentration, all foam morphological properties improved indicating a very minimal effect of CNF aggregation. On the other hand, foam density reduced at low CNF concentrations, but increased at 5 wt.%, and this was attributed to the excessive number of nucleation sites that did not contribute to cell nucleation.

Dlouhá et al. [98] examined the influence of native and surface acetylated CNF (ac-CNF) on the foam morphology of PLA in supercritical foaming. As usual, the cell size decreased by 84% compared with neat PLA foam at 9 wt.% of ac-CNF and the cell density increased from 7.7 × 10^8^ to 5.6 × 10^10^ cells/cm^3^. This was explained in terms of the interplay between the nucleating effect of the CNF surface and the rheological effect on the cell growth. Both the viscosity and melt strength were shown to increase upon addition of CNF. Mi et al. [104] also observed the enhancement of both melt strength and viscosity of PCL upon addition of CNC. The increase of rheological properties, particularly the storage modulus, was greater than that of the loss modulus, suggesting a higher influence of CNC on the elasticity of PCL than on its viscous part. On the other hand, the complex viscosity was also enhanced with the incorporation of CNC particles, as shown in Figure 10. A remarkable increase at 5 wt.% CNC was seen at both high and low frequencies. A dramatic decrease in pore diameter could also be the result of the improved viscoelastic properties of PCL. In addition, the increase of both modulus and complex viscosity could be improved by slight induced crystallinity upon addition of CNC.

The addition of CN particles as a secondary phase into polymer matrices affects their rheological properties and their foaming. The increase of both complex viscosity and storage modulus upon addition of CN was reported. The influence of CN particles on crystallisation also aided in enhancing these properties.

### 7.2. Mechanical Properties

Enhanced crystallisation rates due to nucleation effect of CN particles on improving the degree of crystallinity contribute significantly to the improvement of foam properties. Incorporation of these rigid particles is beneficial in enhancing the mechanical performance of the foam material, even when applied at low amounts. However, the mechanical behaviour of the nanocomposite materials, including foams, is affected by several factors, including the interaction between the polymer matrix and the filler and the amount of filler in the nanocomposite. Although CNs enhance foam properties, it is worth noting that the low strength of bubbles in the polymeric foams may result in the reduced mechanical behaviour of a foam in comparison with an unfoamed material. This kind of observation has been noticed by Lin et al. [100], whereby both flexural strength and modulus of PBS decreased after foaming. However, as compared to neat PBS foam, the incorporation of CNC enhanced these properties because of the synergistic effect of CNC as stress transferring agent and its nucleation capacity in improving effective stress area. However, as shown in Figure 11, 5 wt.% CNC was sufficient to improve both flexural strength and modulus.

At higher concentrations, flexural strength decreased, which was associated with the separation of microphase at higher loadings, leading to a lowered stress transfer and poor mechanical performance. The clustering of CNC particles at higher loadings, as was revealed in SEM images (not included), could be a possible reason for the separation of microphase, therefore, leading to inferior mechanical properties of the foam. On the other hand, a higher concentration of CNC resulted in further increase of flexural modulus, which was ascribed as a possible formation of percolating network from the hydrogen-bonding of CNC. In Table 5, the summarised results on mechanical properties obtained from various CN-biopolymer foams are presented.

Song et al. [118] prepared PVOH/CNC foams crosslinked with formaldehyde at two different crosslinking times (10 and 120 s). The compressive strength at 70% strain was shown to increase from 6.7 to 58.2 KPa and from 65.2 to 114.6 KPa at 10 and 120 s, respectively, with only 1.5 wt.% CNC. This was the maximum CNC concentration for the improvement of compressive strength; a decline of this property was noticed irrespective of initial crosslinking times at higher concentrations. The compressive modulus increased from 7.5 to 43.2 KPa at 10 s with the maximum addition of 1.5 wt.% CNC. However, the incline from 65.1 to 96.6 at 1 wt.% was noticed with followed by a decrease at 1.5 and 2 wt.%. At 1.5 wt.% CNC content and below, the enhancement in both compressive strength and modulus was related to the improved interaction between PVOH and CNC. PVOH and CNC are both hydrophilic—this could therefore be a reason for the improved interaction between two components and better dispersion. However, the decrease of compressive modulus at higher CNC content was attributed to the aggregation of CNC particles and high viscosity of the material, which could be caused by rigidity of CNC particles in restricting mobility of PVOH chains and difficulty in dispersing CNC at high concentrations. Qiu et al. [99] observed the decrease of mechanical properties of PLA/a-CNC foams compared with neat PLA foams. In their work, the decline of tensile properties was associated with several factors, including morphology and interaction between PLA and CNC particles. The Young’s modulus of the nanocomposite foams decreased with the addition of CNC particles because, in the presence of CNC particles, the cell density improved, indicating high volume ratio of pores, hence the observed decrease of the modulus.

Interestingly, the modulus was also shown to be dependent on the interaction of PLA and a-CNC. Depending on the degree of substitution (DS) of CNC, the lower DS resulted in the decrease of the modulus from 202.30 MPa for neat PLA to 128.25 MPa for PLA/CNC foam. However, moderate DS resulted in approximately the same modulus (203.88 MPa) as of neat PLA foams. This observation indicates the effectiveness of stronger interaction between polymer and filler on the mechanical properties of polymer foams. A similar trend was also observed for the tensile strength of the prepared foams. However, a decrease in tensile strength was attributed to the decreased cell wall thickness, as it was observed in the SEM images of nanocomposite foams, which also revealed damages on the cell walls. However, like in case of modulus, moderately substituted CNC showed the increase of tensile strength as compared to neat PLA foam and other composite foams. The dependence of mechanical properties on polymer–filler interaction was also noticed in PLA/CNF foams [120]. The addition of 9 wt.% a-CNF increased the specific modulus by 44% with respect to neat PLA matrix, while the specific strength improved by 46%, possibly because of the increased polymer–filler interaction by acetylation of CNF. However, unmodified CNF also resulted in the enhanced specific modulus and strength, although the results were lower than in modified CNF. This could be due to a very weak interaction between the PLA and unmodified CNF. Improvements in these properties were also obtained at low CNF concentrations (3 wt.%). Interestingly, large increase in specific modulus and strength was noticed in foamed samples. For instance, at the same amount of a-CNF (9%) the specific modulus and strength increased by 19% and 20%, respectively, while in foamed samples 44 and 46% improvement were noticed. Furthermore, the flexibility and toughness of the foams were shown to depend on the fraction of open cells. However, the size of cells appeared to have less or no effect on the tensile properties of foams.

Influence of CNF on mechanical properties of PVOH has also been reported by Liu et al. [119]. In their study, the maximum improvement of all evaluated mechanical properties was noticed at 30 wt.% CNC concentrations. The authors reported 62% improvement in compressive stress at 50% strain of PVOH foam at 30 wt.% CNF content. At higher concentrations, all the investigated mechanical properties, including Young’s modulus, decreased. As explained by the authors, at higher filler content, the macroporous open cells were observed, which can be a cause for poor performance. The formation of macroporous cells was associated with the lack of filaments connection due to the lack of cross-linking by PVOH. However, at concentrations below 30 wt.%, it is believed that amorphous PVOH crosslinks the CNF fibrils together improving the mechanical performance.

Based on the summarised results, the mechanical performance of foams is influenced by the following factors:Matrix–filler interaction—Stronger adhesions results in large improvements in mechanical properties. Modification of CN particles in this regard is the key factor in dictating the compatibility of the two components.Filler concentration—Most of the improvements in mechanical properties occur at CN particle concentration less than 5 wt.%. Higher amounts lead to the weakening of the interfacial area and poor performance.Cell concentration—Increased number of nucleation sites by CN particles results in large cell density; therefore, the presence of a large number of voids results in low modulus.

Optimizations of CN concentrations on CN-based foams are necessary to achieve foam material with balanced properties. Though CN act as a nucleating agent in CN-based foams, it might have a negative influence on reducing the overall mechanical performance of the foam due to a large number of voids created. Consequently, optimum amounts are required to attain a proper structure-property relationship in CN-based foams. Overall, packaging foams with high strength can be developed for commercial purposes, including package cushioning for shock absorption, utilizing CN particles.

### 7.3. Thermo-Mechanical Properties

Thermo-mechanical properties of polymer materials are important for understanding the response of a specific material to an oscillatory deformation in a wide range of temperatures [121]. Dynamic mechanical analysis (DMA) is an effective technique for investigating the response of the material. Molecular transitions occurring as a result of temperature change can be determined. With this technique, the morphology and viscoelastic properties of the semi-crystalline polymer can be determined. The important results that are obtained from this technique are classified into three parameters: flexural modulus (E^′^), loss modulus (E^″^), and tan δ (ratio of E^″^/E^′^). The storage modulus, which represents the tendency of a material to store energy, normally gives qualitative results about the material. In particular, stiffness of the material could be deduced from this parameter. Moreover, E^′^ decreases with the increase of temperature as a result of the increase of polymer chains mobility. This is mostly noticed above T_g_, where the polymer transforms into a rubbery state. The other parameter of interest is the loss modulus, which refers to the ability of a material to dissipate energy; also, it gives an indication about the viscous response of the material. Finally, tan δ provides information regarding the elasticity of the material [3,121].

Thermo-mechanical properties of biodegradable polymeric foams reinforced with CNs have been studied and reported in the literature. The influence of CNC on thermo-mechanical properties of PLA foams has been reported in [97]. The increase of E′ determined in the compressive mode was noticed when CNC was added. At 3% CNC, E′ increased approximately 1.7 times compared with the neat PLA foam at 30 °C. These improvements were associated with the crosslinked structure formed between PLA and CNC, better dispersion and adhesion of CNC and PLA, and enhanced crystallinity. At above 55 °C, the storage modulus increased for all the samples, which was claimed to be due to the densification of the foam materials in the onset of T_g_ of the foam. However, the general trend is that above T_g,_ the storage modulus decreases. The loss modulus of the foams follows the same behaviour. In the tensile mode, E′ of the foams increased approximately 2.2 times with the increase of CNC concentration at 30 °C, which was also related to the crystallinity of the foams. The enhancement of both E′ and E″ was explained by the uniform dispersion of CNC in PLA matrix, which led to the formation of ordered crystalline regions, which subsequently led to the effective transfer of CNC modulus to the PLA. Mi et al. [104] also observed improvements of the storage modulus of PCL foams and solid counterparts when CNC particles were added. However, the storage modulus of foamed PCL samples was much smaller than solid samples because of the presence of voids in the foamed samples. The enhancement of storage modulus of both solids and foams was attributed to the reinforcing effect of rigid CNC particles.

The inclusion of fillers in polymers significantly improves the stiffness of the material. This is in-line with the results reported by several authors [62,66,122,123]. However, in the case of polymer foams, the elasticity is influenced by the presence of cells, which results in a decline of the elasticity of the material. Nevertheless, the inclusion of rigid particles, such as CNC, can compensate for this. Furthermore, the crystallinity of these particles and their effect on the overall crystallinity of the foams also play a critical role in enhancing the stiffness of the foams. Further investigation of the CNF-based foams would be beneficial in determining the effect of CNF, as CNFs are less crystalline than the CNC particles.

### 7.4. Thermal Decomposition

Thermal stability of polymeric foams has been investigated [124,125,126,127,128,129]. The studies have indicated that the incorporation of nanoparticles has a significant effect on the thermal degradation of foams. In addition, the cellular structure of polymer foams has also been shown to influence the degradation of the foam material. For instance, Gepu et al. [127] foamed PLA at different CO_2_ pressures with the solid-state microcellular foaming process. In their work, thermal decomposition temperatures, particularly the onset of degradation (T_d-onset_), were shown to decrease with the increase of foaming pressure. It was explained by the fact that the higher pressure leads to the stronger nucleation weakening the bonding of PLA and enhances cell density and thins the cell walls, which reduces T_d-onset_. Gedler et al. [128] observed a delay on the initial decomposition of polycarbonate (PC) foam compared to the solid PC counterpart. The delay of about 34 °C and 55 °C at 1 and 5 wt.% loss, respectively, was noticed, and it was assigned to the cellular structure of the material, which acted as thermal insulator inhibiting the transfer of heat at initial decomposition stage. However, the rate of mass loss was high in foamed PC, and this was also attributed to the cellular structure, which contributed to internal degradation resulting from the oxidative action of air inside the cellular structure. Furthermore, the studies have also shown the improvement of thermal stability of nanocomposite foams with the addition of nanoparticles [128,129]. Other studies have indicated the much higher increase of the thermal stability, particularly T_d-onset_, of the foamed nanocomposite materials than that of the unfoamed nanocomposite material. The improvements in this regard are attributed to the cellular structure, which acts as a heat insulator during burning and inhibits heat transfer at the onset resulting in a delay of the degradation process and an improvement of thermal stability [128,129,130]. Borkotoky et al. [97] observed single-step degradation of PLA-reinforced CNC foams. With the addition of only 1% CNC, a decrease of T_onset_ by 1.8% was observed. However, at higher loading of CNC (3%), T_onset_ decreased by approximately 7 °C than neat PLA foam. This decline was associated with sulphate concentration during CNC preparation, which reduced the thermal stability of the foam. However, at moderate CNC concentrations (2%), both T_onset,_ and T_max_ were comparable with that of neat PLA foam, which was related to the uniform dispersion of CNC at this concentration.

Generally, CNs are less thermally stable than most synthetic biodegradable polymers. Due to low thermal stabilities, the necessary precautions should be exercised, most importantly in large scale production in industries. Multiple processing steps from CN-composite preparation to continuous foam processes might have an effect on the overall stability of CN and thus deterioration of foam properties. However, this can be controlled by monitoring the processing conditions such as residence time in processing techniques and also the isothermal degradation studies can be useful to predict the residence time. The sulphate groups on CNC also have a significant influence on the thermal stability of CNC and CNC-based composites. As a result, CNF might be a preferred CN material where thermal stability is the key. Overall, well-dispersed CN material lead to better thermal stabilities as it has been shown [97].

### 7.5. Thermal Insulation and Flammability Properties of CN-Based Biodegradable Foams

Although polymer foams are desired for various applications, they should also meet safety requirements. Generally, polymer foams exhibit low thermal conductivity, which makes them good candidates for thermal insulation applications in construction, transportation, and packaging industries. However, polymer foams are easily combustible and highly flammable [131,132]. Consequently, the flammability properties of these materials should be improved to meet specific safety requirements in industries. To our knowledge so far, literature studies based on the flammability of biodegradable polymer foams are still limited.

For the past few decades, halogen-based flame retardant (FR) materials have been mostly used due to low cost and less reduction on mechanical properties of the systems. However, the halogen-based FR materials have been restricted and avoided because of the release of corrosive gases and increased smoke release [133,134]. Regarding health and environmental concerns, environmentally friendly FR materials with a low amount of smoke release are highly desired. The FR materials from renewable materials have gained interest due to the insulating-char-layer formation on the surface of the burning sample. In this regard, cellulose has also attracted interest as a potential material that can be used in the development of friendly FR materials due to its char forming character, however; cellulose is a highly flammable material [133]. Chemical modification of cellulose has been shown to be an effective route to tailor its char forming and improve its flammability properties. The presence of hydroxyl groups on the cellulose backbone allows chemical reactions such as phosphorylation with phosphorus-based compounds which are excellent FR materials. For example, phosphorylation of CNF using ammonium salt of phosphoric acid (NH_4_)_2_HPO_4_ in the presence of urea has been carried out [135]. The flammability properties of nanopaper prepared from phosphorylated CNF were compared with those of filter paper. Upon exposure to flame, filter paper ignited immediately leaving 0% char while the phosphorylated CNF paper exhibited self-extinguishing behaviour when the flame was removed. The cone calorimetry properties of the two materials were also examined under a heat flux of 35 kW/m^2^. The analysis indicated time to ignition (TTI) of 44s and peak heat release rate PHHR of 43 KW/m^2^ for filter paper. Interestingly, modified CNF paper did not ignite under cone calorimetry and this was attributed to the formation of thermally stable char which prohibits the production of volatile components and protecting the uncombusted materials from direct flame contact. Indeed, the cellulose modification with flame-retardant substances provides an alternative way of expanding its applications. Costes et al. [136] showed the influence of cellulose particle size on the flammability properties of cellulose-based composites. The authors observed a reduction in PHRR of PLA/CNC containing 20 wt.% CNC whereas 20 wt.% MCC showed little or no significant influence on PHHR of PLA. The cone calorimeter tests also indicated 36% reduction in TTI for PLA/CNC and 17% reduction for PLA/MCC. The reduction noticed in case of CNC was associated with its nanosize dimensions that increased charring action by allowing higher interfacial interaction with PLA. This observation clearly indicates outstanding performance brought by nanoparticles in reinforcing polymers compared to microparticles with the larger size. Therefore, CN-based biodegradable foams with reduced flammability can be developed using CN particles.

The flammability and thermal insulation properties of cellulose-based foams have also been examined. Guo et al. [137] prepared CNF composite foams containing flame-resistant hydroxyapatite (HAP) nanofibers. The vertical burning test results showed self-extinguishing character of CNF-HAP when the flame was removed whereas CNF foam burned completely within 5 s forming a little char. With an increase in HAP content, the flame spread rate was also reduced and the authors attributed this behaviour to strong flame-resistant property and high thermal stability of HAP. Most importantly, the fire growth rate index (FIGRA) which is a parameter used to evaluate the fire properties of building products was also determined. The CNF-HAP foams showed reduced FIGRA values compared to CNF foam suggesting a suppressed fire hazard quality of CNF-HAP foam material. The thermal insulating properties of the prepared foams were also determined, and the thermal conductivity measurements indicated slight decrease from 40.7 mW/(mK) for neat CNF foam to (38.5–39.1) mW/(mK) for CNF-HAP composite foams. This reduction was associated with unique morphology of the composite since neat HAP has high thermal conductivity than cellulose fibers. Wang et al. [132] also investigated the thermal conductivities of CNC/PVOH composite foams prepared in the presence of 1,2,3,4-butane tetracarboxylic acid (BTCA) as a crosslinking agent. The thermal conductivities of neat CNC and CNC/PVOH composite foams were in a range of 0.036–0.04 Wm^−1^ K^−1^. However, with the inclusion of 25 wt.% BTCA, a 35% reduction in the thermal conductivity of PVOH/CNC foam containing 10 wt.% PVOH was noticed compared to neat CNC foam. At this BTCA concentration, the foam sample exhibited very small cell sizes, which resulted in the lowest thermal conductivity. Foam materials with small cell sizes restrict the mobility of gas molecules and thus the reduction in heat transfer through the gaseous phase. The prepared foams at this concentration showed low thermal conductivity (0.027 Wm^−1^ K^−1^) compared to that of commercial expanded polystyrene (0.030–0.040 Wm^−1^ K^−1^). The vertical burning test showed a retained structural integrity with a minimal shrinkage after burning for composite foam while CNC foam collapsed after 11s of burning. Though cellulose composite foams show improvement in both thermal insulation and flammability properties, Zhou and co-workers [138] did not observe any significant change on the thermal conductivities and flammability properties of CNF modified tannin-furanic resin foams compared to neat CNF foam. Wicklein et al. [139] prepared nanocomposites foam consisting of graphene oxide (GO), sepiolite nanorods (SEP) and boric acid as crosslinking agent at various concentration with respect to CNF. The nanocomposites foam containing 10 wt.% GO and 10 wt.% SEP showed a reduction in thermal conductivities in radial orientation from 18 mW^−1^ K^−1^ (CNF foam) to 15 mW^−1^ K^−1^ (composite foam). The reduction was attributed to thermal properties of the nanomaterials and the foam structure. From an industrial perspective, the design and application of renewable cellulose-based foams with thermal insulation properties in the range of commercially used PS and PU foams can be beneficial in construction and building applications. However, the inferior flammability properties of cellulose-based foams can be tailored by the inclusion of FR substances or chemical modifications of CN with FR materials. With these strategies, high-performance CN-based biodegradable incombustible foams can be attained which would also be important for packaging, construction, and building industries.

## 8. Application of CN-Biopolymer Foams

Although studies have dwelt into the development of CN-biodegradable nanocomposite materials and their foaming behaviour, the specific applications of these materials have not been explored that much. Very few studies have evaluated the wettability of CN-biopolymer foams, which is a characteristic property in biomedical applications [50,97,104]. Wettability determines the level of hydrophobicity/hydrophilicity of the foam material, which gives an indication about the surface energies of the material. It is known that highly hydrophilic materials possess very high surface energy, which is desired for cell adhesion in tissue engineering. Morouço et al. [50] observed an increase of surface energies of PCL/CNF scaffolds after addition of CNF. With the addition of CNF particles, a decline in contact angles was noticed; it is an indication of the increase of wettability of the material, which is beneficial for cell adhesion. In the other study [104], the performance of PCL/CNC foams for tissue engineering scaffolds application was investigated. The tests were carried out by 3T3 fibroblasts for 10 days. It was observed from the fluorescence images that PCL and its nanocomposites up to 1% CNC showed mostly live cells, which is an indication of good compatibility of the PCL/CNC foam material with the cells. However, in this case, the performance of the foamed material was also shown to be dependent on CNC concentration.

## 9. Biodegradation of CN-Based Biopolymer Foams

Biodegradation of polymer materials has been reported [140,141,142]. The biodegradation of polymer foams has also been evaluated, although the information is still insufficient. There are certain factors that inherently affect the biodegradation of polymer materials. These include the molecular weight of the polymer, the nature of the material (amorphous/semi-crystalline polymer), the polymer architecture, and the level of hydrophobicity of the material [140,141]. However, in the case of polymer foams, biodegradation is also dependent on the structure of the cells. For instance, in closed-cell foams, the surface area exposed to biodegrading agents is minimised, which retards or prolongs the biodegradation of the material. In an open-cell structure, the surface area that can be attacked by the microorganism is high, and, as a result, biodegradation can occur effectively within shorter times [142]. The biodegradation of PVOH/CNF (60 wt.%) foam has been evaluated in compost, which is similar to a real-life biodegrading environment, for 30 days [119]. The cellular structure of the foams was destroyed after this period. SEM images (cross-sectional) indicated a high accumulation of microbes on the PVOH/CNF foams, probably due to the high susceptibility of CNF towards microbial attack. The obtained results show that CN particles might contribute to enhancing the biodegradability of the foam material because of their high susceptibility toward microorganisms in natural environments.

## 10. Conclusions

Because of the issues associated with the non-environmental friendliness and biodegradability of most widely used conventional polymers, it is necessary to develop materials that are compatible with the environment. Conventional polymer foams derived from petroleum oil–based polyolefins are among the polymer materials that significantly contribute to plastic pollution. To overcome this issue, studies have indicated that biodegradable polymers can be used as an alternative to conventional polymers. However, several challenges remain regarding biodegradable polymers. First, cost of biodegradable polymers in comparison with conventional polymers restricts their utilisation in most fields of application. Second, their processing, specifically for foaming, is a challenge, especially for polymers such as PLA. Nevertheless, from this review, we have seen that the performance and processing of biodegradable-based foams can be enhanced by incorporating cellulose nanoparticles. Cellulose nanoparticles were shown to facilitate the foaming of polymers, resulting in improved rheological properties and crystallisation behaviour. Improvements in the mechanical, thermal, and dynamic properties upon the incorporation of these nanoparticles at very low concentrations (~1–5 wt.%) clearly indicated the effect of cellulose nanoparticles in improving the foaming process and, at the same time, acting as a reinforcing material. Furthermore, they are biodegradable, environmentally friendly, and abundant, as they can be obtained from natural resources. They can replace most inorganic nanoparticles which are toxic and not environmentally friendly. From the findings summarised in this paper, CN-based biodegradable nanocomposite foams are an alternative for various applications, including packaging, thermal insulation, and other applications where biodegradability of the material is required.

## Figures and Tables

**Figure 1 polymers-11-01270-f001:**
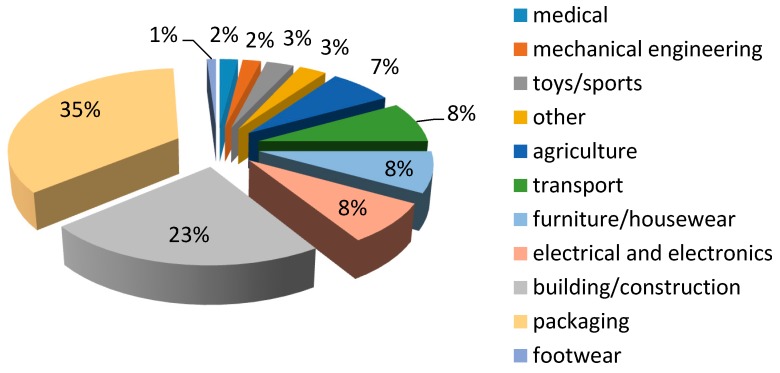
Percentage usage of plastics in various sectors [9].

**Figure 2 polymers-11-01270-f002:**
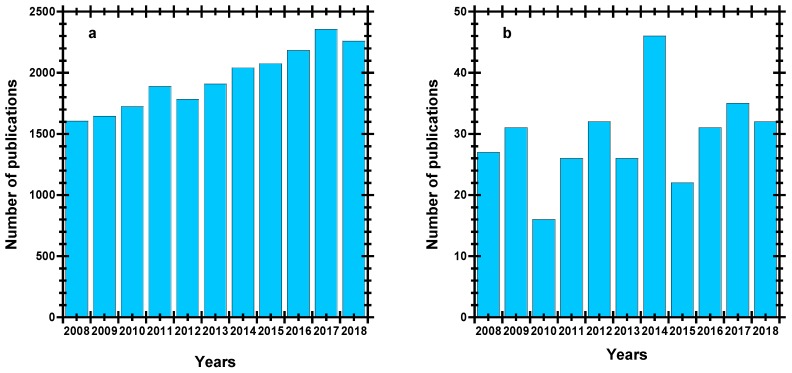
(**a**) Publication history on biodegradable polymers; (**b**) publication history on biodegradable polymer foams from 2008 to 2018 (information obtained from Scopus) (Keywords: biodegradable polymers and biodegradable polymer foams).

**Figure 3 polymers-11-01270-f003:**
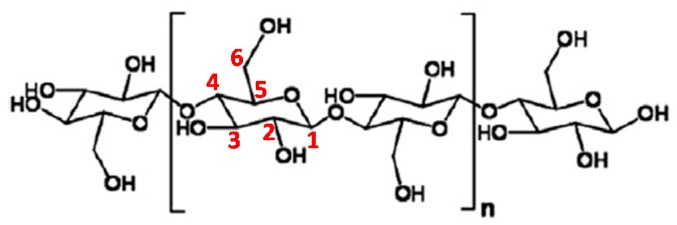
Molecular structure of cellulose.

**Figure 4 polymers-11-01270-f004:**
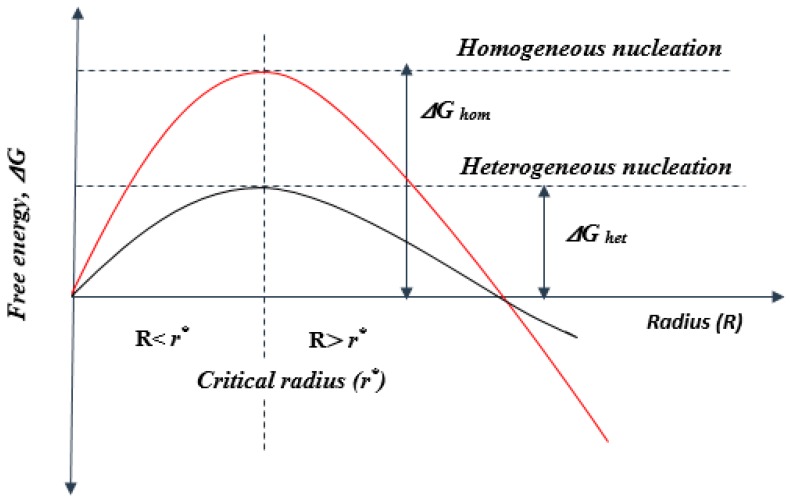
Bubble nucleation and growth as a function of free energy.

**Figure 5 polymers-11-01270-f005:**
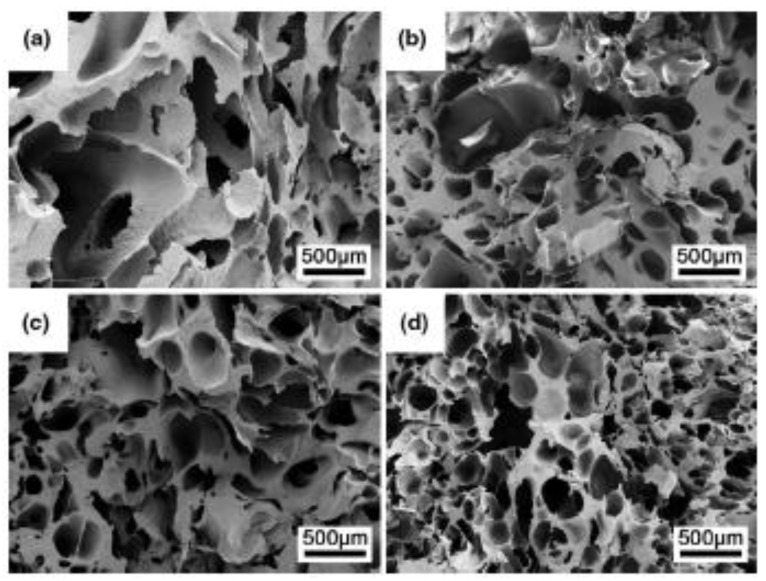
Scanning electron microscopy (SEM) images of injection molded foams: (**a**) PCL, (**b**) 0.5 % CNC, (**c**) 1 % CNC, (**d**) 5 % CNC [104]. Reproduced with permission from Springer Nature. Copyright © 2014.

**Figure 6 polymers-11-01270-f006:**
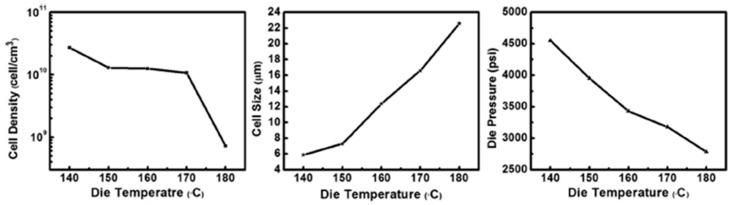
Cell density, cell size, and die pressure at different die temperatures [107]. Reproduced with permission from American Chemical Society (ACS). Copyright © 2014.

**Figure 7 polymers-11-01270-f007:**
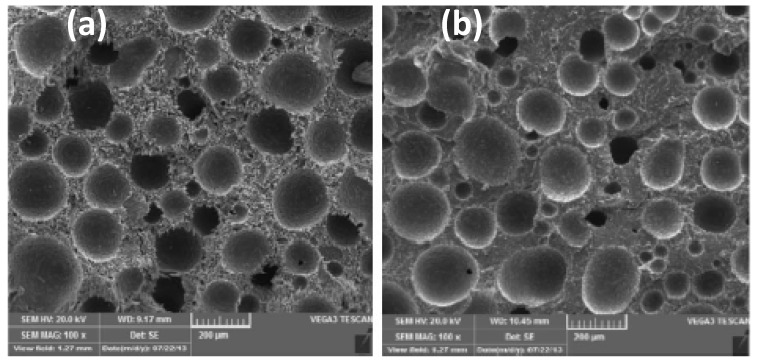
SEM images of the cross-section of the foamed composites: (**a**) PBS/AC (5%); (**b**) PBS/CNC (3%)/AC (5%); (**c**) PBS/CNC (5%)/AC (5%); (**d**) PBS/CNC (10%)/AC (5%) [100]. Reproduced with permission from Springer Nature. Copyright © 2015.

**Figure 8 polymers-11-01270-f008:**
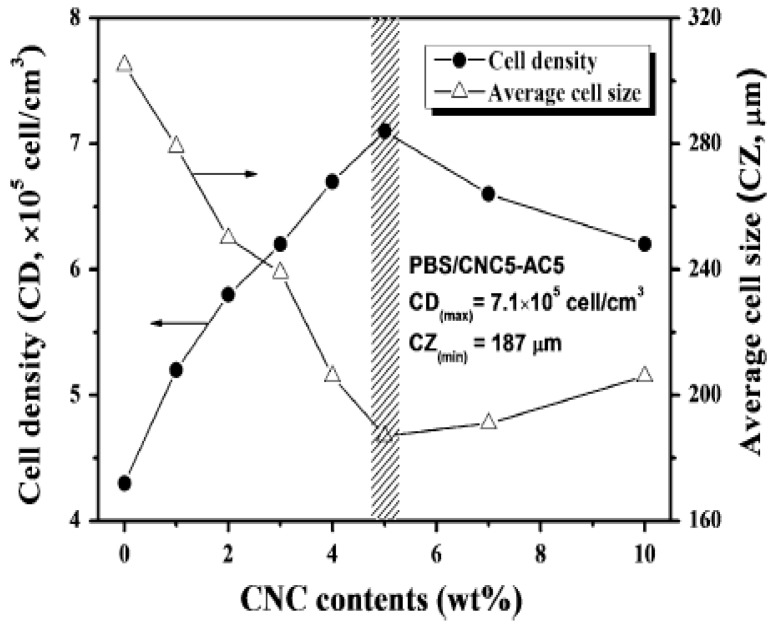
Measured cell density and average cell size as a function of CNC concentration in PBS/CNC foams [105]. Reproduced with permission from Springer Nature. Copyright © 2015.

**Figure 9 polymers-11-01270-f009:**
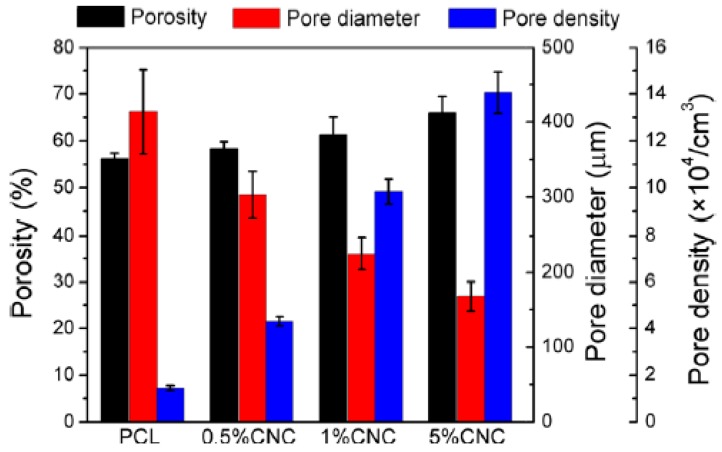
Porosity, pore diameter, and pore density of PCL/CNC foams at various CNC concentrations [104]. Reproduced with permission from Springer Nature. Copyright © 2014.

**Figure 10 polymers-11-01270-f010:**
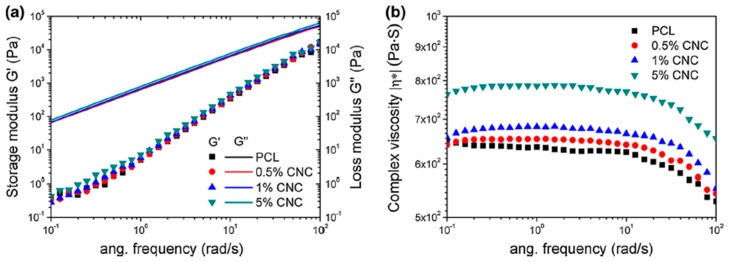
The rheological results of PCL and PCL/CNC nanocomposites: (**a**) storage modulus and (**b**) complex viscosity as a function of angular frequency [104]. Reproduced with permission from Springer. Nature. Copyright © 2014.

**Figure 11 polymers-11-01270-f011:**
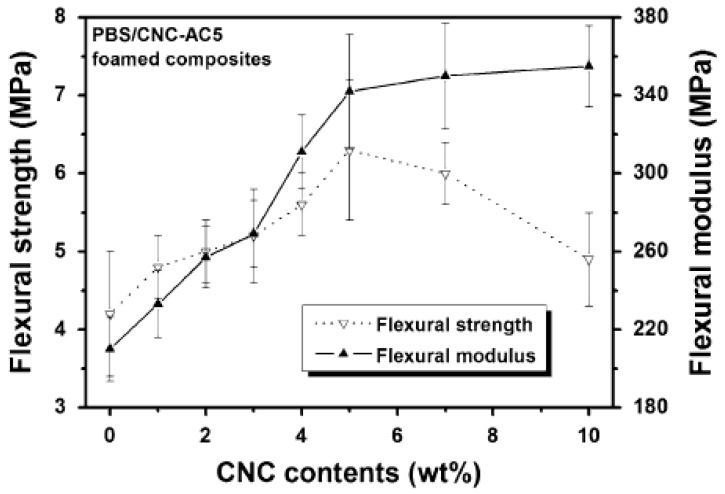
Effect of CNC concentration on flexural strength and modulus [100]. Reproduced with permission from Springer Nature. Copyright © 2015.

**Table 1 polymers-11-01270-t001:** Classification of biodegradable polymers.

Origin	Family	Common Examples	Refs.
Biomass	Polysaccharides	Cellulose, Starch, Chitin, etc.	[2,30]
	Proteins	Collagen, Gelatin, Albumin, Soya, Glutan	
Microorganisms	Polyhydroxyalkanoates (PHA)	Polyhydroxybutyrate (PHB)	[2,29,30]
		Poly (γ-glutamic acid)	
		Poly (hydrobutyrate-*co*-hydroxyvalerate) (PHBV)	
Petroleum oil	Polyesters	Poly (ԑ-caprolactone) (PCL)	[29,30,33]
		Poly (butylene succinate) (PBS)	
		Poly (butylene succinate-*co-*adipate) (PBSA)	
		Poly (butylene adipate-*co*-terephthalate) (PBAT)	
Bio-derived	Polyesters	Poly (lactic acid) (PLA)	[30]

**Table 2 polymers-11-01270-t002:** Summary of methodologies used to synthesise cellulose nanostructure (CN)-based nanocomposites.

CN-Based Nanocomposite	CN Modification	Processing Strategy	Solvent	Temp./°C	Time	Screw Speed/rpm	Refs.
PLA/CNF	-	Solvent evaporation	Water	-		-	[49]
PCL/CNF	-	Solvent evaporation	DMF ^#^	-		-	[50]
PLA/CNF	-	Solvent evaporation	dichloromethane	-	-	-	[51]
		Melt mixing/batch	-	170	10 min	60	
PLA/CNF	acetylation	Solvent casting	chloroform	-	60 min	-	[52]
PLA/PHBV/CNC	TEMPO-mediated oxidation	Solvent casting	chloroform			-	[53]
PHBV/CNF		Solvent casting	chloroform/DMF	50	30 h	-	[54]
PLA/CNF	Esterification by oleic acid	Solvent casting	chloroform			-	[55]
PLA/PBS/CNC	Surfactant-modified	Solvent casting	chloroform	-		-	[56]
PCL/CNC	acetylation	Solvent casting	chloroform	-		-	[57]
PLA/CNC	Acetylation	Solvent casting	dichloromethane	-		-	[23]
PCL-MFC	Grafting polymerisation	-	Toluene	100		-	[58]
PLA/CNC	PEG ^$^-grafted-CNC	Solution-based (electrospun)	DMF/chloroform	ambient		-	[59]
PBS/MFC	CNC-modified by acetylchloride with ball milling	Solvent-based (MB ^§^)	DMF	60		-	[28]
		Melt-mixing/batch	-	140	5 min	60	
PBAT/CNC	Octadecyl isocyanate	Melt mixing (batch)	-	130	3 min	30	[60]
PBS/PLA/CNC	CNC-grafted-PBS	Melt mixing/batch	-	190	10 min	60	[61]
PHBV/CNC	-	Melt mixing/small-extrusion	-	165			[26]
PBAT/CNC	acetylation	Melt mixing/batch	-	120	10 min	50	[27]
PBSA/CNC	-	Melt mixing/batch	-	120	10 min	50	[62]
PBAT/CNC	PBG *-grafted-CNC	Melt extrusion	-	150	5 min	100	[44]
PLA/CNC	Silanisation	Melt extrusion/batch	-	165	5 min	100	[63]
PHB/CNC	PLA-grafted-CNC	Melt extrusion/batch	-	180	5 min	50	[64]
PLA/PCL/CNC	CNC-grafting-PCL & PLA	Melt extrusion/batch	-	165	5 min	100	[65]
PLA/CNC	-	Melt mixing/batch	-	190	10	60	[66]
PCL/CNC	CNC-grafted-PBMA ^¥^; Micelles; Latex	Melt mixing	-	110	6 min	100	[67]

* PBG-Polybutyl glutarate; ^§^ MB-master batch; ^#^ DMF-N, N-Dimethylmethanamide; ^$^ PEG-Polyethylene glycol; ^¥^ PBMA-Poly (n-butyl methacrylate).

**Table 3 polymers-11-01270-t003:** Processing strategies for the fabrication of biodegradable foams.

Origin	Biopolymer-Based Foam	Processing Strategy	Blow Agent	Refs.
Biomass	Tannin/furfuryl alcohol	Solvent-based	Self-blowing	[77]
	Chitin hydrogels	High-pressure cell	ScCO_2_	[78]
	Albumin protein	Solvent based & Microwave drying	Self-blowing	[79]
	Cassava starch containing additives	Baking process	-	[80]
	Thermoplastic starch coated with chitosan	Baking process	Water	[81]
	CMCNa/PEGDA700 ^1^	Solvent-based	Surfactant (Pluronic)	[82]
Microorganisms	PHBV/Clay	Extrusion foaming	ScCO_2_	[83]
	PHBV	Extrusion foaming	Sodium bicarbonate & citric acid	[84]
	PHB/Natural Fibre	Batch Process (pressure quench)	CO_2_	[85]
Bio-derived	PLA/cellulosic fiber	Injection foaming	N_2_ (0.5 wt.%)	[86]
	PLA	Injection foaming	N_2_	[87]
	PLA	Extrusion foaming	ScCO_2_	[88]
	PGA	Batch process	ScCO_2_	[89]
	PLA/PBSA	Injection foaming	N_2_	[15]
	PLA/PBAT	Extrusion foaming	ScCO_2_	[90]
Petroleum-oil	PCL	Batch-process	ScCO_2_	[91]
	PCL/HA ^2^	Batch process	ScCO_2_	[92]
	PBAT/MTPS ^3^	Single-screw extruder	SAFTEC^®^ UBA-60 ^5^ (3, 5, 7 wt.%)	[93]
	PBS	Mold method	Ammonium bicarbonate (1–10 wt.%)	[94]
	PBS/TMPTMA/DCP ^4^	Compression molding	Azodicarbonamide	[95]

^1^ CMCNa/PEGDA-Sodium salt of carboxymethylcellulose and polyethylene glycol diacrylate; ^2^ HA-Hydroxyapatite; ^3^ MTPS-Maleated Thermoplastic Starch; ^4^ TMPTMA/DCP-Trimethylolpropane trimethy/dicumyl peroxide; ^5^ SAFTEC^®^ UBA-60-.

**Table 4 polymers-11-01270-t004:** Summary of cell size and density of PLA/CNC foams [97].

Sample	Average Cell Diameter (μm)	Cell Density × 10^5^ (mm^−3^)
Neat PLA	2.8 ± 1.3	2.1
PLA/CNC-1%	1.04 ± 0.43	3.0
PLA/CNC-2%	1.5 ± 0.54	2.7
PLA/CNC-3%	1.81 ± 0.61	2.9

**Table 5 polymers-11-01270-t005:** Mechanical properties of various CN-based biopolymer foams.

CN-Based Foam	CN-Concentration	Mechanical Property	% Improvement	Refs.
PVOH/CNC-crosslinked with formaldehyde for 10 and 120 s	1.5 wt.%	Compressive strength at 70% strain (KPa)	769% at 10 s 76% at 120 s	[118]
		Compressive modulus (KPa)	476% at 10 s 9% at 120 s 60% at 120 s *	
PVOH/CNC	30 wt.%	Compressive stress at 50% strain (KPa)	62%	[119]
		Young’s modulus (MPa)	75%	
		Energy absorbed (KJ/m^3^)	35%	
PLA/CNF	9 wt.%	Specific modulus	44%	[120]
		Specific strength	46%	

* Maximum compressive modulus obtained at 0.5 wt.% CNC.

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
