# Peer review of "Cellulose Nanostructure-Based Biodegradable Nanocomposite Foams: A Brief Overview on the Recent Advancements and Perspectives"

_polymers, 2019, doi:10.3390/polym11081270_

Round 1
Reviewer 1 Report
none, only English should be improved
Author Response
Thank you for your much-appreciated effort and time. Your recommendation has been taken care in revised version.
Reviewer 2 Report
This work reports a critical and accessible overview of the influence of cellulose nanostructure (CN) particles on the properties of biodegradable foams; in particular, their rheological, thermal, and mechanical properties and biodegradability. From the findings summarized in this paper, CN-based biodegradable nanocomposite foams are an alternative for various applications, including packaging, thermal insulation, and other applications where biodegradability of the material is required. The work is informative to the field of CN. I believe this manuscript is acceptable. However, the authors should consider the following points to further improve the manuscript.
1. Lines 109-110:Cellulose can be obtained from various sources, such as plants, animals, bacteria, and algae. In fact, animals are not a source of cellulose.
2. Author must combine Figure 9.
3. Figures 8 and 12 are not very clear. Please improve resolution of the Figures.
4. Lines 532-533:at a higher percentage, the cell density started to decrease, while the cell size increased, please explain.
5. Lines689. “the incline from 65.1 to 96.61”, please keep the decimal point consistent as the scientific reports.
6. At last, the level of English throughout this manuscript should be improved. There are a number of grammatical errors and instances of badly worded/constructed sentences. Please check the manuscript and refine the language carefully.
Author Response
Firstly, accept our appreciation for the meticulous assessment of the manuscript. We really thank you because your comments and recommendations helped to improve the quality of our work. We have carefully considered the recommendations and the following changes/additions have been made.
Response 1: Thank you for this comment. This has been removed in revised version.
Response 2: Thank you for this recommendation. It has been taken into account in revised version and new Figure number is 7.
Response 3: On the basis of your recommendation, the resolution of mentioned figures has been improved.
Response 4: Thank you for this suggestion which has been corrected accordingly.
Response 5: Thank you for your comment. The decimal point has been corrected.
Response 7: Thank you for much appreciated comment. The language has been refined throughout the manuscript.
Reviewer 3 Report
The manuscript reviews recent advancements in biodegradable nanocomposite foams from cellulosic materials. Tough it is a short review, it covers different important perspectives of the subject matter effectively, and this would attract readers of similar interest. However, a moderate improvement is required before consideration. Specific comments as follows:
1. There is some very basic information provided, which are not necessary and do not add any value to this review. For example, Section 4 – Polymer foams and their classification section is way too general. Figure 4 can be found in a relevant book. I would suggest removing them and add a few topics on real-time applications of CN-based composites.
2. Figure 6 is very too general. Add some case studies relevant to the injection foaming.
3. I suggest adding flammability properties of CN foams under ‘7. Properties of cellulose-nanostructured nanocomposite foams’. This area is a hot topic and should not be ignored in such a review.
4. Should add some progressive graphical representation on biodegradability of CN-based biopolymer foams.
5. Every section is short, and they read well. However, it should benefit the readers to relate with most recent applications, instead of just describing what they are. Especially, given the high number of references being cited, the manuscript can be more detailed.
6. Most importantly, critical analysis is missing. What are the pro and cons of each aspect discussed?
7. Page 13 - Why are those sentences underlined with increased font size?

Author Response
Firstly, accept our appreciation for the meticulous assessment of the manuscript. We really thank you because your comments and recommendations helped to improve the quality of our work. We have carefully considered the recommendations and the following changes/additions have been made.
Response 1: We fully agree with you. Section 4, including Figure 4 have been removed and they are replaced with the flammability and thermal insulation properties of CN-based foams.
Response 2: On the basis of your recommendation, the figure 6 has been revised and corrected accordingly.
Response 3: Thank you for your comment. A brief overview on flammability properties of CN-based foams has been given. Besides, an overview on thermal insulation properties has been covered.
Response 4: Thank you for your comment. Since the information on biopolymer foams is limited, the progressive graphical representation on CN-based foams, whereby the images showing biodegradation as a function of time are shown, is not available. A detailed study on biodegradation of biopolymer foams is crucial as it would aid in understanding the influence of foam structures on biodegradation rates. Therefore, more studies on biodegradation kinetics of biopolymer foams would be helpful.
Response 5: Thank you for your comment. The properties of CN-based foams have been linked with the possible applications as shown under section 6.
Response 6: Thank you for your comment. A general conclusion is given at the end of each concept discussed.
Response 7: Thank you for your question. The authors found it difficult to notice the underlined sentences in page 13
Thank you again for your much-appreciated effort and time. Finally, we hope you will be satisfied with our revision and that the manuscript will now be acceptable for publication in Polymers as a review article.
Round 2
Reviewer 3 Report
The authors have revised the manuscript with new added discussion on the 'thermal insulation and flammability properties of CN-based biodegradable foams' as per the suggestions. The manuscript reads well. A few corrections will be necessary:
Figure 3 - Indicate the red numbers in the captions
Wt% should be wt.% - It is not consistently given.
Use of capital letter should be corrected, especially in headings and subheadings.